# Anti-Inflammatory Effect of Synaptamide in Ischemic Acute Kidney Injury and the Role of G-Protein-Coupled Receptor 110

**DOI:** 10.3390/ijms25031500

**Published:** 2024-01-25

**Authors:** Anna A. Brezgunova, Nadezda V. Andrianova, Aleena A. Saidova, Daria M. Potashnikova, Polina A. Abramicheva, Vasily N. Manskikh, Sofia S. Mariasina, Irina B. Pevzner, Ljubava D. Zorova, Igor V. Manzhulo, Dmitry B. Zorov, Egor Y. Plotnikov

**Affiliations:** 1A.N. Belozersky Institute of Physico-Chemical Biology, Lomonosov Moscow State University, 119992 Moscow, Russia; brezgunova.anna.2014@post.bio.msu.ru (A.A.B.); andrianova@belozersky.msu.ru (N.V.A.); abramicheva.polina@belozersky.msu.ru (P.A.A.); manskikh@mail.ru (V.N.M.); pevzner_ib@belozersky.msu.ru (I.B.P.); ljuzor@belozersky.msu.ru (L.D.Z.); zorov@belozersky.msu.ru (D.B.Z.); 2Faculty of Bioengineering and Bioinformatics, Lomonosov Moscow State University, 119992 Moscow, Russia; 3Faculty of Biology, Lomonosov Moscow State University, 119991 Moscow, Russia; aleena.saidova@gmail.com (A.A.S.); dpotashnikova@gmail.com (D.M.P.); 4Department of Chemistry, Lomonosov Moscow State University, 119991 Moscow, Russia; sofia.mariasina@yandex.ru; 5Faculty of Fundamental Medicine, Lomonosov Moscow State University, 119991 Moscow, Russia; 6Research and Educational Resource Center “Pharmacy”, RUDN University, 117198 Moscow, Russia; 7V.I. Kulakov National Medical Research Center for Obstetrics, Gynecology and Perinatology, Ministry of Healthcare of Russian Federation, 117198 Moscow, Russia; 8A.V. Zhirmunsky National Scientific Center of Marine Biology, Far Eastern Branch, Russian Academy of Sciences, 690041 Vladivostok, Russia; i-manzhulo@bk.ru

**Keywords:** kidney, ischemia/reperfusion, acute kidney injury, inflammation, synaptamide, therapy, nephroprotection

## Abstract

The development of drugs for the treatment of acute kidney injury (AKI) that could suppress the excessive inflammatory response in damaged kidneys is an important clinical challenge. Recently, synaptamide (N-docosahexaenoylethanolamine) has been shown to exert anti-inflammatory and neurogenic properties. The aim of this study was to investigate the anti-inflammatory effect of synaptamide in ischemic AKI. For this purpose, we analyzed the expression of inflammatory mediators and the infiltration of different leukocyte populations into the kidney after injury, evaluated the expression of the putative synaptamide receptor G-protein-coupled receptor 110 (GPR110), and isolated a population of CD11b/c^+^ cells mainly representing neutrophils and macrophages using cell sorting. We also evaluated the severity of AKI during synaptamide therapy and the serum metabolic profile. We demonstrated that synaptamide reduced the level of pro-inflammatory interleukins and the expression of integrin CD11a in kidney tissue after injury. We found that the administration of synaptamide increased the expression of its receptor GPR110 in both total kidney tissue and renal CD11b/c^+^ cells that was associated with the reduced production of pro-inflammatory interleukins in these cells. Thus, we demonstrated that synaptamide therapy mitigates the inflammatory response in kidney tissue during ischemic AKI, which can be achieved through GPR110 signaling in neutrophils and a reduction in these cells’ pro-inflammatory interleukin production.

## 1. Introduction

Currently, there are no effective biomedical approaches to treating such a widespread clinical syndrome as acute kidney injury (AKI). This disease affects up to 15% of all hospitalized patients, while its occurrence has been reported for more than 50% of patients staying in intensive care units [1]. Basically, the treatment of this disease is limited to various types of hemodynamic procedures and renal replacement therapy [2]. One of the main pathological mechanisms of renal tissue damage in ischemic AKI is the development of an excessive inflammatory response [3]. In this context, targeting inflammation is considered a promising method for the treatment of AKI of various etiologies [4].

An inflammatory response is triggered in the kidney tissue very shortly after ischemic injury [5]. Neutrophils, monocytes/macrophages, dendritic cells (DCs), and natural killers (NKs) serve as a source of cytokines such as interleukins and tumor necrosis factor-alpha (TNF-α), as well as other important intermediates that affect AKI and cause damage to kidney tissue. Damage-associated molecular patterns (DAMPs) and alarmins derived from necrotic renal cells activate resident immune cells, parenchymal cells, and the endothelium, which attract circulating immune cells [6]. The most pronounced response comes from neutrophils, which can cause further damage to epithelial tubular cells [7]. Neutrophils produce proteases, myeloperoxidase, reactive oxygen species, and cytokines, leading to increased vascular permeability and decreased integrity in the tubular epithelial and endothelial cells, exacerbating renal damage [8].

Several drugs are currently being developed that could mitigate inflammation in kidney tissue in the early and late stages of AKI. One of the promising compounds that has been shown to reduce inflammation in various experimental models is the endogenous metabolite and structural analog of anandamide, synaptamide (N-docosahexaenoylethanolamine) [9]. Synaptamide has been shown to have a broad spectrum of beneficial effects, including neuroprotective, anti-inflammatory, and neurogenic properties, making it a promising therapeutic agent for various neurological diseases and injuries [10]. The possible mechanisms of synaptamide’s anti-inflammatory effects are thought to be realized through a reduction in the levels of arachidonic acid-derived eicosanoids, its conversion to anti-inflammatory lipid metabolites [11], or signaling through G-protein-coupled receptor 110 (GPR110) [12].

The aim of our study was to investigate the therapeutic efficacy of synaptamide in ischemia/reperfusion (I/R)-induced AKI and to evaluate the possible molecular mechanisms of synaptamide’s effects. Given the anti-inflammatory effects of synaptamide, we investigated the levels of pro-inflammatory cytokines in kidney tissue 48 h after injury with synaptamide treatment. One of the aims of the study was to assess the infiltration of immune cells and to determine which type of immune cells infiltrate the damaged tissue the most. We are also trying to identify possible mechanisms by which the therapeutic effect of synaptamide can be achieved in AKI. In addition, we investigated the changes in the metabolic profile in the serum of rats after synaptamide therapy.

## 2. Results

### 2.1. Synaptamide Mitigates Inflammation in Kidney Tissue after Ischemic Injury

Since the inflammatory response is high after renal I/R, we investigated the effects of synaptamide treatment on a number of inflammatory mediators (Figure 1). We showed that the levels of some cytokines were increased in kidney tissue 48 h after I/R, including interleukin (IL) 1 beta (IL-1β), tumor necrosis factor α (TNF-α), and IL-6 (Figure 1A–F). Synaptamide administration significantly reduced the expression of IL-1β and transforming growth factor β1 (TGFβ1) (Figure 1A,B) and tended to reduce the levels of TNF-α and IL-6 48 h after I/R (Figure 1C,D). In addition, we observed an upward trend in the mRNA expression of inducible nitric oxide synthase (iNOS) (Figure 1G), cyclooxygenase-2 (COX-2) (Figure 1H), and complement component 3 (C3) (Appendix A) in the kidneys after I/R, while synaptamide treatment decreased COX-2 expression. In addition, we analyzed the expression of anti-inflammatory IL-10, which decreased in kidney tissue after I/R, but to a lesser extent in rats administered with synaptamide (Appendix A).

### 2.2. The Effects of Synaptamide on Leukocyte Infiltration in Kidney Tissue after Ischemic Injury

During renal I/R, we investigated the presence of different leukocyte populations and the effect of synaptamide on their abundance (Figure 2). First, we examined the expression of cluster of differentiation (CD) 11a (CD11a), an integrin that regulates cellular adhesion and is present on all leukocytes [13]. We demonstrated that the mRNA expression of CD11a in the kidney increased significantly after I/R, whereas the administration of synaptamide reduced its expression (Figure 2A). Another pan-leukocyte marker, CD45, tended to increase after I/R in both the untreated and synaptamide-treated groups (Figure 2B). In addition, we examined the expression of the neutrophil marker chemokine (C-X-C motif) ligand 1 (CXCL1) and observed a statistically significant increase in both the “I/R” and “I/R+synaptamide” groups (Figure 2C). We also evaluated the expression of the specific macrophage marker CD68, which increased after I/R (Figure 2D). Although synaptamide did not change the total amount of macrophages, the therapy reduced the number of CD86^+^ cells (Figure 2E), which are presumed to be M1 macrophages, and increased the amount of CD163^+^ cells, which are M2 macrophages (Figure 2F) [14].

Since neutrophils are the major immune cells infiltrating kidney tissue as a result of I/R, we assessed the amount of neutrophils on kidney slices (Figure 3A–C). We demonstrated that I/R led to a significant increase in the number of neutrophils in the kidney, which was partially attenuated by the administration of synaptamide during I/R (Figure 3D). Of note, we observed hyaline cylinders in the tubules, the desquamation of epithelial cells, and mitotic phase cells in the tubules after I/R in both experimental groups.

We also assessed leukocyte infiltration using flow cytometry on kidney cells stained with antibodies against CD11b/c (Figure 4A–C), which mainly reveals neutrophils and macrophages [13]. We demonstrated that both I/R and I/R with synaptamide treatment resulted in a significant increase in CD11b/c^+^ cells compared to the control group (Figure 4D).

### 2.3. Synaptamide Exerts Its Anti-Inflammatory Effects through GPR110 in Neutrophils

Recently, it has been shown in other experimental models that synaptamide can exert its anti-inflammatory effect via GPR110 in immune cells [12]. In this regard, we explored the expression of this receptor in cells obtained from the kidneys and demonstrated that synaptamide significantly increased GPR110 expression during renal I/R, which was not upregulated in rats treated with synaptamide without AKI induction (Figure 5A).

To analyze which cells might be responsible for such an increase in GPR110 expression, we obtained CD11b/c-positive (CD11b/c^+^) cells, which are mainly neutrophil and macrophage populations [15]. We demonstrated that the mRNA expression of GPR110 was increased in CD11b/c^+^ cells from the kidneys of rats exposed to I/R with synaptamide treatment (Figure 5B). We also examined the expression of pro-inflammatory markers TNF-α and IL-1β in CD11b/c^+^ cells, which tended to decrease after I/R with synaptamide administration (Figure 5C,D).

In addition, we examined the mRNA expression of GPR110 and TNF-α in peripheral blood mononuclear cells (PBMCs). We found that GPR110 was not expressed in PBMCs, in contrast to CD11b/c^+^ cells from kidney tissue (Figure 5E). Of note, GPR110 mRNA expression was not activated in PBMCs 48 h after I/R or synaptamide treatment. Furthermore, synaptamide had no effect on TNF-α expression in these cells after I/R (Figure 5F). Based on the data obtained, we hypothesize that the anti-inflammatory effects of synaptamide are mediated by GPR110 in CD11b/c^+^ cells, which are presented mainly by neutrophils and macrophages. Since the majority of cells infiltrating the kidney tissue after renal I/R are neutrophils, the main effect of synaptamide may be in modulation of the inflammatory response of these exact cells.

### 2.4. Effects of Synaptamide on AKI Severity

We evaluated the severity of AKI without or with synaptamide treatment (Figure 6). We found that blood urea nitrogen (BUN) and serum creatinine (SCr) concentrations increased significantly 48 h after renal I/R. However, synaptamide treatment did not mitigate kidney dysfunction despite its anti-inflammatory effect (Figure 6A,B). We also analyzed a more sensitive marker of kidney damage, neutrophil gelatinase-associated lipocalin (NGAL), in urine samples 24 h after I/R (Figure 6C). We observed a significant increase in NGAL levels in the urine of rats from the “I/R” and “I/R+synaptamide” groups. We also evaluated the expression of another sensitive biomarker, kidney injury molecule-1 (KIM-1), in the kidney tissue 48 h after injury and demonstrated that the mRNA expression of KIM-1 was significantly increased with I/R and tended to decrease in the kidneys of rats treated with synaptamide during I/R.

### 2.5. Synaptamide Affects Serum Metabolic Profile after Ischemic Injury

We assessed the impact of I/R and synaptamide treatment on a panel of serum metabolites (Figure 7). We observed that 3-hydroxybutyrate, methionine, and cytidine increased after I/R and tended to decrease in rats with synaptamide administration. At the same time, 2-hydroxyisovalerate, isobutyrate, 3-hydroxyisovalerate, 2-Hydroxyisobutyrate, N,N-dimethylamine, N,N-dimethylglycine, choline, hippurate, myoinositol, tartrate, trigonelline, mannose, 2′-deoxyuridine, creatinine, and trans-aconitate significantly increased after I/R, but remained unaltered after synaptamide administration. Some metabolites (O-acetylcarnitine, 3-indoxylsulfate, creatine, p-toluenesulfonic acid, trimethylamine N-oxide (TMAO), allantoin, N-phenylacetylglycine, glucuronate,) demonstrated a significant increase in the “I/R+synaptamide” group compared to “I/R”. Synaptamide administration significantly decreased threonine and citrate levels compared to I/R alone and tended to reduce valine and formate concentrations. The level of tryptophan in the serum decreased after I/R, and this decrease stayed at the same level after the administration of synaptamide. Of note, we observed I/R- or synaptamide-induced changes in the serum levels of several unidentified substances (Figure 7), which were labeled as d109, s274, s275, t429, s277, s343, s356, d507, d659, d576, s233, t355, s218, s219, t290, and d455-d456-d477 (see Section 4).

## 3. Discussion

Various drugs are currently being developed for the treatment of AKI that could suppress the excessive inflammatory reaction in the kidney tissue after damage. There are several potential therapeutic approaches to treat AKI that target components of the inflammatory cascades [16]. However, the existing anti-inflammatory strategies have detrimental limitations. In particular, non-steroidal anti-inflammatory drugs (NSAIDs), which are commonly used to treat inflammatory responses by inhibiting COX-2, lead to a decrease in prostaglandin E2 (PGE2) levels, particularly in the kidney tissue [17]. In response to the reduced PGE2 level, a decrease in blood flow in the afferent arterioles of the glomeruli and a compensatory increase in vasopressin in blood are observed, which can lead to vasoconstriction that further impairs the blood supply to the renal tissue and eventually causes pre-renal AKI [18]. Thus, the use of classical NSAIDs, which demonstrate a high ratio of COX-1/COX-2 inhibition, is not recommended in cases of kidney injury [18].

In this context, it is necessary to develop drugs that target alternative pathogenic pathways of the inflammatory process during AKI. Synaptamide (N-docosahexaenoylethanolamine), the structural analog of anandamide, the endogenous ligand of the cannabinoid receptor, has been proposed as a promising therapeutic agent [19]. In general, omega-3 and omega-6 fatty acid endocannabinoids have been shown to be metabolized via lipoxygenase (LOX)-, COX- and cytochrome-P450 (CYP450)-mediated pathways. Synaptamide, which is synthesized from docosahexaenoic acid, has been shown to be metabolized only by LOX and CYP450 [20]. In addition, it has been hypothesized that synaptamide may indirectly reduce the pro-inflammatory response by reducing arachidonic acid-derived COX eicosanoids, or it may be further converted by eicosanoid-synthesizing enzymes to form anti-inflammatory lipid metabolites [11].

The aim of this study was to explore the effects of synaptamide on inflammation under AKI induced by renal I/R; this has not been investigated before. It is known that in ischemic AKI, the cells of the immune system are activated very rapidly during the reperfusion phase [3]. During renal I/R, cells of innate immunity, e.g., neutrophils, monocytes/macrophages, DCs, NK cells, and natural T killer cells, which are not antigen-specific, play key roles in the development of inflammation. Neutrophils attach to the activated endothelium and accumulate in the kidney, which has been demonstrated in both animal models and human AKI [7,21,22,23], particularly in the peritubular capillary network of the outer medulla, as early as 30 min after reperfusion. It has been shown that the number of macrophages in the kidney increases as early as 1 h after reperfusion, reaches a peak after 24 h, and persists for 7 days [24].

Synaptamide has previously been shown to have a broad spectrum of anti-inflammatory, neuroprotective, and neurogenic properties, making it a promising therapeutic agent for various neurological diseases and injuries. For example, treatment with synaptamide in traumatic brain injury or sciatic nerve chronic constriction injury decreased astrogliosis and immunoreactivity and reduced the levels of inflammatory biomarkers such as glial fibrillary acidic protein (GFAP), S100β, and IL-6 [25,26,27,28]. Synaptamide prevented a lipopolysaccharide (LPS)-mediated increase in the production of the pro-inflammatory cytokines TNF-α, IL-1β, and IL-6 in microglial cells [29]. In our study, for the first time, we demonstrated that the administration of synaptamide resulted in a remarkable anti-inflammatory effect during ischemic AKI. Synaptamide treatment during renal I/R reduced the expression of some pro-inflammatory mediators such as IL-1β and TGFβ1 (Figure 1), suggesting that synaptamide can modulate specific inflammatory pathways.

In addition, we found that synaptamide treatment caused an increase in anti-inflammatory IL-10 levels in renal tissue after I/R (Appendix A). This suggests that synaptamide may modulate both pro- and anti-inflammatory factors within the renal inflammatory response. The same tendencies were observed for neural tissue after injury. For instance, synaptamide administration reversed the LPS-mediated increase in IL-1β and TNF-α production in the hippocampus [29] and modulated pro-inflammatory (IL-1β, IL-6) and anti-inflammatory (IL-4, IL-10) cytokine levels in serum and the spinal cord after chronic constriction injury [30].

In our study, we investigated the infiltration of different leukocyte populations in renal tissue after renal I/R and showed that neutrophils were recruited the most when considering the multiple increases in *CXCL1* expression (Figure 2C). We also quantified neutrophils on renal slices and found a significant increase in these immune cells after I/R, which was partially attenuated via the administration of synaptamide during I/R (Figure 3). The content of macrophages, as measured by the expression of *CD68*, also increased 48 h after renal I/R (Figure 2D). Interestingly, the administration of synaptamide decreased the leukocyte integrin CD11a, which was elevated after I/R. Previously, anti-CD11a therapy indicated a beneficial effect on renal function after I/R and resulted in smaller tubular necrosis and lesser leukocyte infiltration after injury [31].

Recent studies have shown that synaptamide modulates the presence of different types of macrophages. During nerve injury, synaptamide increased the number of CD163^+^ cells associated with anti-inflammatory M2 macrophages and decreased the number of CD68^+^ macrophages, indicating a reduction in pro-inflammatory M1 macrophages [27]. Similarly, synaptamide demonstrated anti-inflammatory effects by decreasing the expression of pro-inflammatory markers (CD86 and MHC II) and promoting the production of anti-inflammatory markers (Arg1 and CD206) and downregulating the activity of ionized calcium-binding adapter molecule 1 (Iba-1)-positive macrophages [29]. Synaptamide showed the ability to increase CD206 synthesis in macrophage cell culture after LPS-induced inflammation [30]. In our study, although the total amount of macrophages remained unaffected by synaptamide, the therapy significantly decreased the content of CD86^+^ cells (presumably M1 macrophages) and increased CD163^+^ cells (M2 macrophages) (Figure 2D–F), suggesting a modulation of macrophage polarization.

To better understand the molecular mechanisms of synaptamide’s effect, we investigated the role of GPR110, which was hypothesized to be its receptor [32]. Unfortunately, due to the lack of specific inhibitors and activators of GPR110, we cannot perform inhibition analysis and clearly determine the role of GPR110 in synaptamide signaling pathways. However, there is some indirect evidence that signaling through this receptor is involved in the anti-inflammatory effect of synaptamide in AKI. Indeed, we detected a significant increase in GPR110 expression in the renal tissue of rats treated with synaptamide (Figure 5A), indicating the possible involvement of GPR110 in mediating the effects of synaptamide in renal AKI. Interestingly, the expression of this receptor did not increase in intact animals receiving synaptamide (Figure 5A). To identify the specific cell populations responsible for the increased GPR110 expression, we isolated CD11b/c^+^ cells, which are mainly composed of neutrophils and macrophages [15]. Indeed, CD11b/c^+^ cells from the kidneys of rats subjected to I/R treatment with synaptamide demonstrated increased GPR110 expression (Figure 5B). Moreover, CD11b/c^+^ cells showed a decreased expression of pro-inflammatory markers such as TNF-α and IL-1β after synaptamide administration (Figure 5C,D). At the same time, we found that GPR110 was not expressed in PBMCs, which are mainly presented by lymphocytes and monocytes [33], both in intact rats and those exposed to I/R (Figure 5E), and that synaptamide did not affect the expression of pro-inflammatory cytokines in these cells (Figure 5F). These results lead us to hypothesize that the anti-inflammatory effect of synaptamide is likely realized via GPR110-mediated pathways in neutrophils (Figure 8), which was first demonstrated in the context of AKI.

Despite the demonstrated anti-inflammatory effect of synaptamide in renal I/R, the drug was unable to improve renal function at selected doses and durations (Figure 6). This may be due to the fact that the pathophysiology of AKI involves complex cellular and molecular signaling pathways that include inflammation, oxidative stress, apoptosis, and microvascular dysfunction [34]. Modification of the synaptamide treatment protocol may be necessary to achieve a reduction in the severity of ischemic kidney injury. To gain valuable insight into the systemic effects of I/R injury and synaptamide treatment, we analyzed the levels of selected metabolites in serum (Figure 7). Most of the metabolites detected can be attributed to the uremic toxins that retain in the blood during renal dysfunction [35]. Synaptamide administration tended to modulate ketone body and amino acid metabolism and nucleotide biosynthesis. Overall, the observed effects of synaptamide on specific metabolites underscored the potential of synaptamide as a modulator of metabolic pathways associated with renal damage.

## 4. Materials and Methods

### 4.1. Animals

The study was performed on male 3-4-month-old Wistar outbred rats (*n* = 29). The animal protocols were reviewed and approved by the Animal Ethics Committee of the A.N. Belozersky Institute of Physico-Chemical Biology Lomonosov Moscow State University: Protocol 3/19 dated March 18, 2019. All procedures were performed in accordance with the «Animal Research: Reporting of In Vivo Experiments» (ARRIVE) guidelines. Animals had unlimited access to food and water and were kept in cages in a temperature-controlled environment (20 ± 1 °C) under the 12 h/12 h light/dark regime.

### 4.2. Renal I/R

Rats were anesthetized and placed on a thermostatically controlled heating pad to maintain body temperature at 37 ± 0.5 °C during the procedure. The vascular bundle of the left kidney was clamped with a microvascular clip for 40 min leading to the halt of kidney blood flow. After 40 min of ischemia, blood flow was reinstated to the left kidney by removing the microvascular clip. Prior to restoring circulation by removing the clip, the nephrectomy of the contralateral kidney was performed. Blood and kidney tissue samples were harvested 48 h after I/R injury for further analysis; urine samples were collected 24 h after I/R. A subset of rats underwent subcutaneous injection of synaptamide at the dose of 10 mg/kg for 5 days prior to the I/R, on the day of procedure (1 h before I/R), and 1 day post I/R (Figure 9) [36]. Synaptamide was obtained as previously described [27].

### 4.3. Flow Cytometry and Cell Sorting

Single-cell suspensions were prepared from the kidneys of intact rats, 48 h after I/R, and I/R with synaptamide treatment, using MediMachine (BD, Franklin Lakes, NJ, USA), and passed through a 40 μm cell filter. For cell surface staining, the monoclonal antibodies anti-CD11b/c-PE (Miltenyi Biotec, Bergisch Gladbach, Germany) were used. For this, the cells were incubated with the antibodies at +4 °C for 20 min, washed with phosphate buffered saline (PBS), and analyzed using a FACSAria SORP cell sorter (BD Biosciences, Franklin Lakes, NJ, USA). Diva software v.6.1.0 (BD Biosciences, Franklin Lakes, NJ, USA) was employed for the data analysis. CD11b/c^+^ cells were sorted using a FACSAria SORP cell sorter (BD Biosciences, Franklin Lakes, NJ, USA) with a 70 μm nozzle and corresponding pressure parameters.

### 4.4. Real-Time PCR

For the quantitative real-time PCR, we collected samples of kidney tissue, blood, or CD11b/c^+^ cells obtained via cell sorting.

Kidney tissue was homogenized in an appropriate volume of Trizol reagent (Thermo Fisher Scientific Inc., Invitrogen, Waltham, MA, USA) using Potter homogenizer. After isolation of the aqueous phase containing nucleic acids using chloroform and treatment with 75% ethanol, further steps of total RNA isolation and treatment with DNase were performed using RNeasy plus mini kit (QIAGEN GmbH, Hilden, Germany), following the protocol recommended by the manufacturer. After RNA extraction, the concentration and purity were determined by using the NanoPhotometer (Implen, München, Germany) to measure the RNA concentration and purity according to А260/А280 and А260/А230 ratios. 

RT-PCR experiments were performed using CFX96 Touch cycler (Bio-Rad, Hercules, CA, USA) with SYBR Green I (Bio-Rad, USA). All samples were processed in triplicate. Primers (DNA-synthesis, Russia) used in this study (Table 1) were designed with the Beacon Designer 7 program (Premier Biosoft Int., San Francisco, CA, USA). Primer specificity was confirmed via melting curve analysis. The primer efficiencies were calculated by generating a standard curve for each target gene using a five-fold serial dilution of the cDNA pool and were in the range of 1.8–2.0. mRNA expression levels were calculated as E^−ΔCt^, where E is the primer efficiency and Ct is the cycle number at which the product fluorescence rose above the threshold level. These values were normalized to the threshold cycles of hypoxanthine phosphoribosyltransferase 1 (*HPRT*), polyubiquitin-C (*UBC*)*,* or 60S acidic ribosomal protein P0 (*RPLP0*).

Blood was collected from an anesthetized rat through the jugular vein using a syringe preloaded with 1000 IU of heparin solution. The obtained blood sample was then layered onto a Ficoll solution and subjected to centrifugation at 600 g for 30 min. Following centrifugation, the serum was aspirated using a vacuum system, and the PBMC-enriched ring was meticulously collected using a pipette and subsequently transferred into a 10-fold volume of RLT buffer. RNA extraction was performed following the standard RNEasy protocol for animal tissues.

### 4.5. Serum Analysis

Blood samples were taken from the carotid artery. After 15 min hold at room temperature, the clot was removed via centrifugation at 2000× *g* for 5 min at 2–4 °C. The resulting serum was analyzed for SCr and BUN using the AU480 Chemistry System (Beckman Coulter, Brea, CA, USA).

### 4.6. Western Blotting

The urine samples were centrifuged at 10,000× *g* for 5 min, mixed with 2× sample buffer (12 μL of sample and 12 μL of 2× sample buffer) containing 10% 2-mercaptoethanol, and boiled for 5 min. Kidneys were homogenized in 5 mL PBS with 1 mM of protease inhibitor phenylmethylsulfonyl fluoride (PMSF), then centrifuged at 1000× *g* for 3 min. Protein concentration was measured using the bicinchoninic acid kit (Sigma Aldrich, St. Louis, MO, USA). Prior to electrophoresis, samples were centrifuged at 10,000× *g* for 5 min and loaded onto 15% Tris-glycine polyacrylamide gels (20 μL to each lane of polyacrylamide gel in the case of urine samples and 10 μg protein per lane in the case of kidney samples).

After electrophoresis, proteins were transferred onto PVDF membranes (Amersham Pharmacia Biotech, Amersham, UK). Membranes were blocked with 5% non-fat milk in PBS with 0.05% Tween-20 and subsequently incubated with primary rabbit antibodies against NGAL (Abcam, Cambridge, UK), primary rabbit antibodies against IL-6 (Thermo Fisher Scientific, Waltham, MA, USA), or primary mouse antibodies against TNFα (Sigma, USA). Membranes were then incubated with secondary anti-rabbit IgG or anti-mouse antibodies conjugated with horseradish peroxidase (Jackson ImmunoResearch, West Grove, PA, USA) and probed with Advansta Western Bright ECL kit (Advansta, San Jose, CA, USA). Detection was performed using V3 Western Blot Imager (BioRad, Hercules, CA, USA). Western blots of kidney homogenates were normalized to the intensities of the same samples on stain-free images (Appendix A). 

### 4.7. Histology

Kidneys fixed in paraformaldehyde were embedded in paraffin and sectioned into 4 μm slices. H&E staining was performed by immersing sections in alum hematoxylin for 5 min, followed by a 5 min wash in running tap water, and subsequent staining in 1% eosin B for 10 min. All sections were dehydrated by passaging through alcohols, cleared in xylene, and then mounted. The stained kidney slices were examined using an Axio Scope A1 microscope (Carl Zeiss, Jena, Germany) equipped with an MRc.5 camera (Carl Zeiss, Germany).

### 4.8. Quantification of Serum Metabolite Level

Quantification of metabolite level in rat serum was performed using nuclear magnetic resonance (NMR) spectroscopy. Samples were prepared according to the recommendations in [37]. Serum was mixed with methanol and chloroform, precooled to −20 °C, vortexed 1500 rpm for 30 min at 6 °C, kept for 3 h at −20 °C, and then centrifuged for 14 min at 12,000× *g* and 5 °C. The upper water–methanol layer was carefully separated and dried using a SpeedVac concentrator for 14 h at room temperature until the solvents completely evaporated. The samples were stored at −60 °C until measurements were taken.

The dried extracts were dissolved in a deuterated phosphate buffer (20 mM, pH 7.1) containing 2 mM sodium azide and d6-DSS as an internal standard. After vortexing for 5 min at room temperature and centrifugation for 5 min, the samples were placed into standard 5 mm NMR tubes. The spectra were acquired on AVANCE Neo 700 MHz NMR spectrometer (Bruker BioSpin, Ettlingen, Germany) equipped with a Prodigy triple resonance probe (Bruker, Germany) at 25 °C, using the noesypr1d pulse sequence with the following parameters: 131,072 data points, 4 dummy scans, 64 scans, 19.8364 ppm spectral width, 4.7 s accumulation time, and 3.0 s relaxation delay between scans.

Metabolite identification was performed using the Chenomx v9 program based on the built-in database and the data from the Animal Metabolite Database (AMDB) [38]. We also observed several unidentified substances, which were labeled as d109, s274, s275, t429, s277, s343, s356, d507, d659, d576, s233, t355, s218, s219, t290, and d455-d456-d477, where s/d/t represents multiplicity of observed NMR signal (single, double, triplet), numbers correspond to its chemical shift multiplied by 100, and symbol “-” separates different signals of the one unidentified substance. The signal-to-noise ratio for the DSS signal (100 µM) was estimated as 443 ± 92, which allowed us to obtain the concentrations of other metabolites using the DSS signal as an internal reference. The concentrations of metabolites were measured related to the DSS peak area using the integration option in Mestrenova v10.0.2 program. Hierarchical clustering analysis of serum metabolites was performed using Heatmapper (http://heatmapper.ca/, accessed on 22 November 2023). The Euclidean distance algorithm for similarity measurement and the complete linkage clustering algorithm were selected. For each parameter, the decimal logarithm of the fold change was used in comparison to the control.

### 4.9. Statistical Analysis

The statistical data analysis was performed using the GraphPad Prism 7 software (GraphPad Software Inc., La Jolla, CA, USA). The data were tested for normal distribution using the Shapiro–Wilk test. For comparisons among multiple groups, ANOVA with Tukey’s multiple comparison test was used in cases of parametric variables and Kruskal–Wallis test with Dunn’s multiple comparisons test was used in cases of nonparametric variables. *p* < 0.05 was admitted statistically significant. The values within a group are expressed as means ± SEM.

## 5. Conclusions

In this study, for the first time, the anti-inflammatory effect of synaptamide, the structural analog of anandamide, the endogenous ligand of a cannabinoid receptor, was demonstrated in a model of ischemic AKI. We showed that treatment with synaptamide reduced the levels of the proinflammatory cytokines IL-1β and TGFβ1 in kidney tissue after injury, which was due to a decrease in their expression in the immune cells infiltrating the kidney tissue. We assume that the observed anti-inflammatory effects are mediated by GPR110 receptors in neutrophils. However, despite the pronounced anti-inflammatory effect, synaptamide treatment failed to improve renal function. These results may be due to multiple mechanisms of AKI development, which simultaneously trigger inflammation, oxidative stress, apoptosis, and microvascular dysfunction, while synaptamide does not affect all of these processes. We hope that synaptamide or its derivatives can be used in therapy for AKI or other diseases.

## Figures and Tables

**Figure 1 ijms-25-01500-f001:**
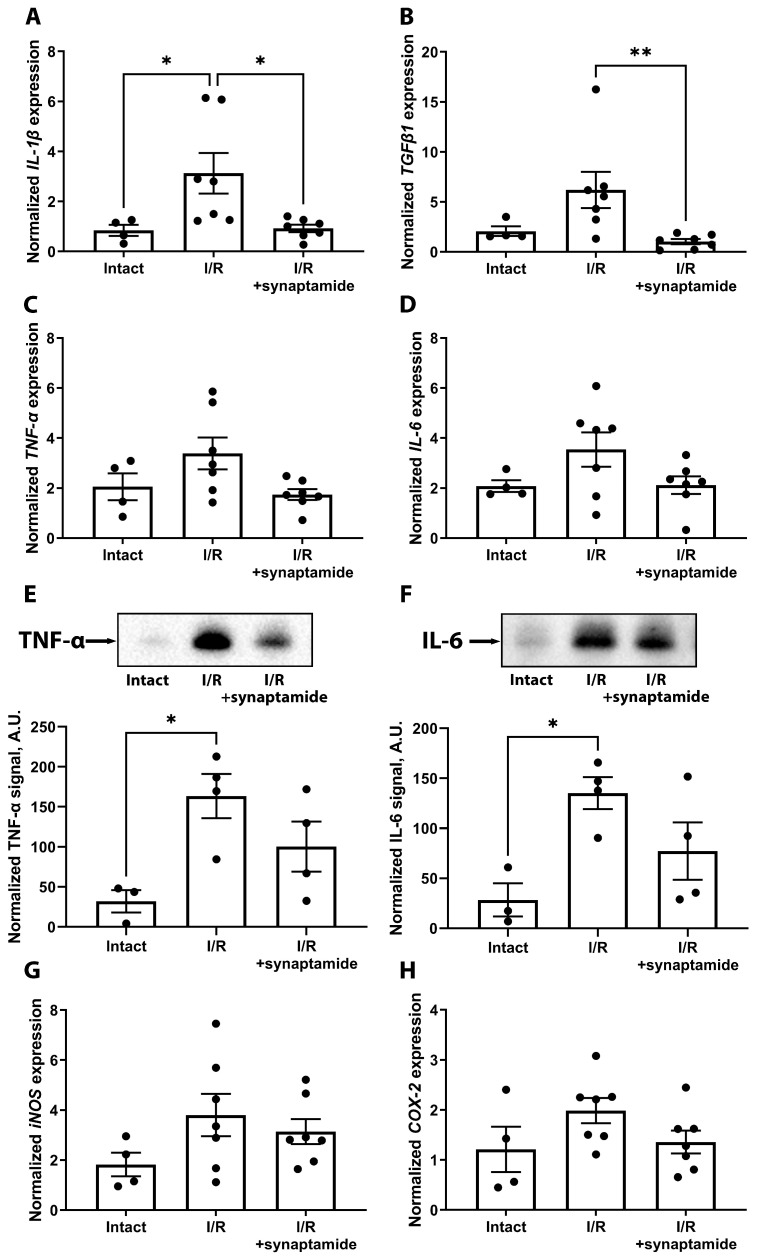
The levels of pro-inflammatory mediators after renal I/R and synaptamide treatment. (**A**) *IL-1β* mRNA expression. (**B**) *TGFβ1* mRNA expression. (**C**) *TNF-α* mRNA expression. (**D**) *IL-6* mRNA expression. (**E**) Normalized TNF-α level measured via Western blotting. (**F**) Normalized IL-6 level measured via Western blotting. (**G**) *iNOS* mRNA expression. (**H**) *COX-2* mRNA expression. * *p* < 0.05, ** *p* < 0.01 (one-way ANOVA with Tukey’s multiple comparison test or Kruskal–Wallis test with Dunn’s multiple comparisons).

**Figure 2 ijms-25-01500-f002:**
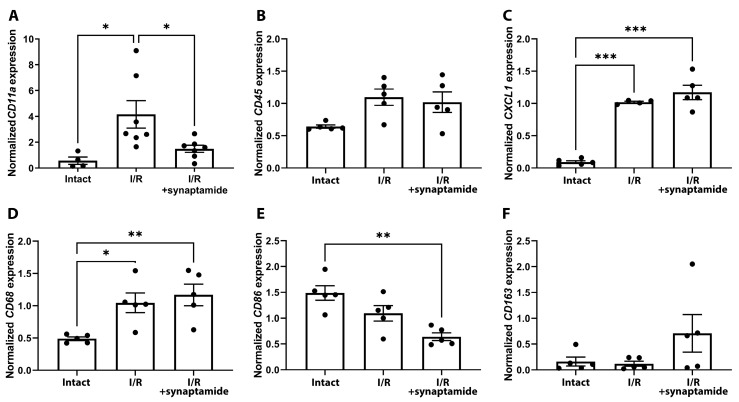
The appearance of various leukocyte populations in the kidney tissue after renal I/R and synaptamide treatment. (**A**) *CD11a* mRNA expression. (**B**) *CD45* mRNA expression. (**C**) *CXCL1* mRNA expression. (**D**) *CD68* mRNA expression. (**E**) *CD86* mRNA expression. (**F**) *CD163* mRNA expression. * *p* < 0.05, ** *p* < 0.01, *** *p* < 0.001 (one-way ANOVA with Tukey’s multiple comparison test).

**Figure 3 ijms-25-01500-f003:**
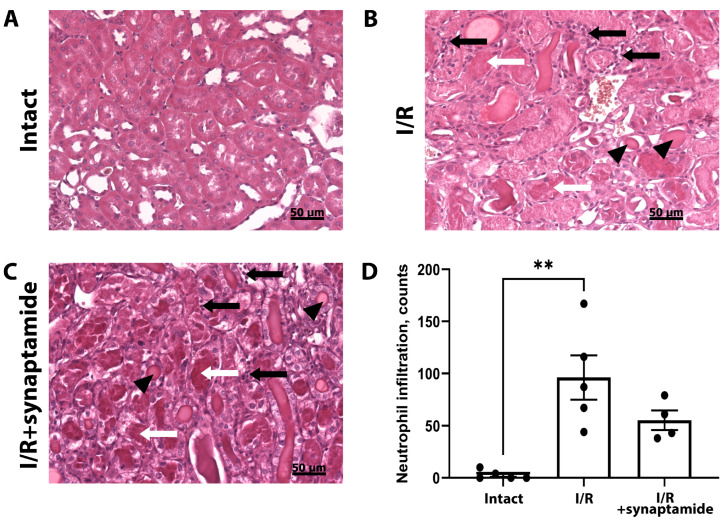
Neutrophil infiltration in kidney tissue after renal I/R and synaptamide treatment. H&E staining of kidney sections of intact rats (**A**), 48 h after I/R (**B**), or 48 h after I/R with synaptamide treatment (**C**). Neutrophils (black arrows), hyaline cylinders (black arrowheads), and the desquamation of epithelial cells (white arrows) are observed in kidney tissue after renal I/R. (**D**) Estimation of the number of neutrophils in kidney tissue of intact rats, 48 h after I/R or 48 h after I/R with synaptamide treatment. ** *p* < 0.01 (Kruskal–Wallis test with Dunn’s multiple comparisons).

**Figure 4 ijms-25-01500-f004:**
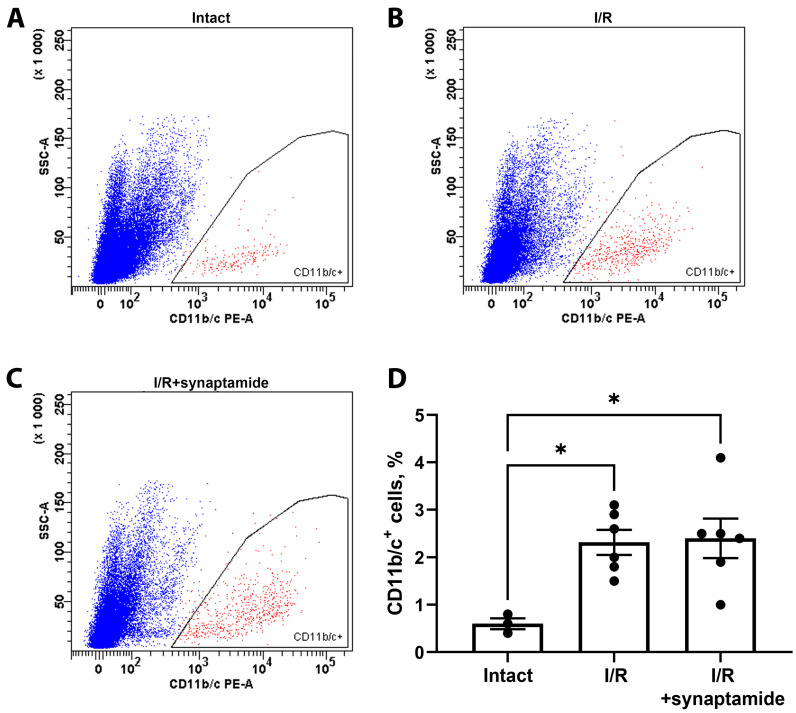
Infiltration of CD11b/c^+^ cells in kidney tissue after renal I/R and synaptamide treatment. Flow cytometry of cells obtained from the kidneys of intact rats (**A**), 48 h after I/R (**B**), or 48 h after I/R with synaptamide treatment (**C**), stained with fluorescent-labeled antibodies against CD11b/c^+^. (**D**) The content of CD11b/c^+^ cells in the kidneys of intact rats, 48 h after I/R, or 48 h after I/R with synaptamide treatment. * *p* < 0.05 (one-way ANOVA with Tukey’s multiple comparison test).

**Figure 5 ijms-25-01500-f005:**
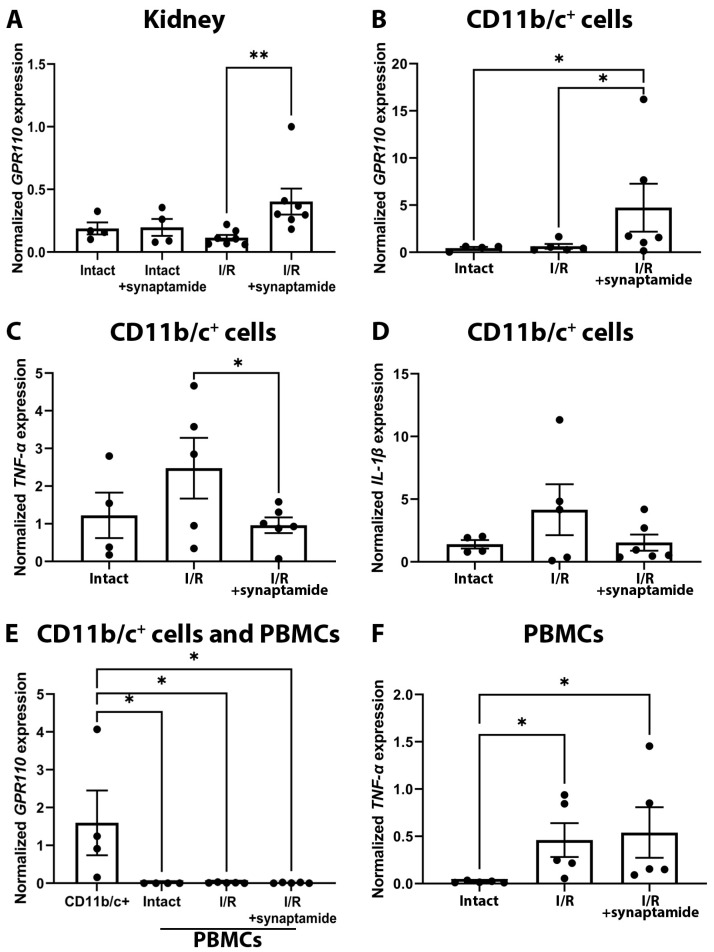
GPR110-associated anti-inflammatory effects of synaptamide in renal I/R. (**A**) *GPR110* mRNA expression in whole kidney tissue. (**B**) *GPR110* mRNA expression in CD11b/c^+^ cells from kidney tissue. (**С**) *TNF-α* mRNA expression in CD11b/c^+^ cells from kidney tissue. (**D**) *IL-1β* mRNA expression in CD11b/c^+^ cells from kidney tissue. (**E**) Comparison of *GPR110* mRNA expression in CD11b/c^+^ cells from intact kidney tissue and PMBCs obtained from the blood of intact rats, 48 h after I/R, or 48 h after I/R with synaptamide treatment. (**F**) *TNF-α* mRNA expression in PBMCs. * *p* < 0.05, ** *p* < 0.01 (one-way ANOVA with Tukey’s multiple comparison test or Kruskal–Wallis test with Dunn’s multiple comparisons).

**Figure 6 ijms-25-01500-f006:**
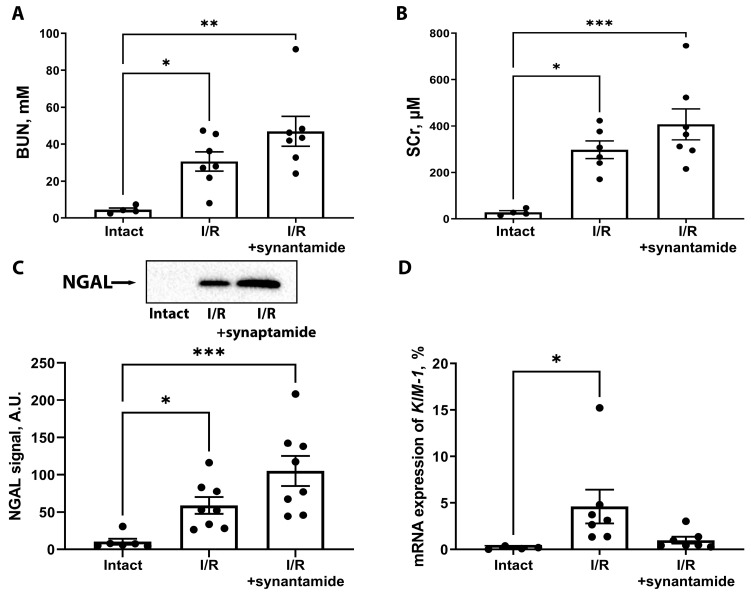
The severity of AKI in rats after I/R with or without synaptamide treatment. (**A**) BUN levels in intact rats, 48 h after I/R, and I/R with synaptamide treatment. (**B**) SCr concentration 48 h after I/R and I/R with synaptamide treatment. (**C**) Urinary NGAL levels 24 h after I/R and I/R with synaptamide treatment evaluated via Western blotting. (**D**) KIM-1 mRNA expression 48 h after I/R and I/R with synaptamide treatment. * *p* < 0.05, ** *p* < 0.01, *** *p* < 0.01 (one-way ANOVA with Tukey’s multiple comparison test or Kruskal–Wallis test with Dunn’s multiple comparisons).

**Figure 7 ijms-25-01500-f007:**
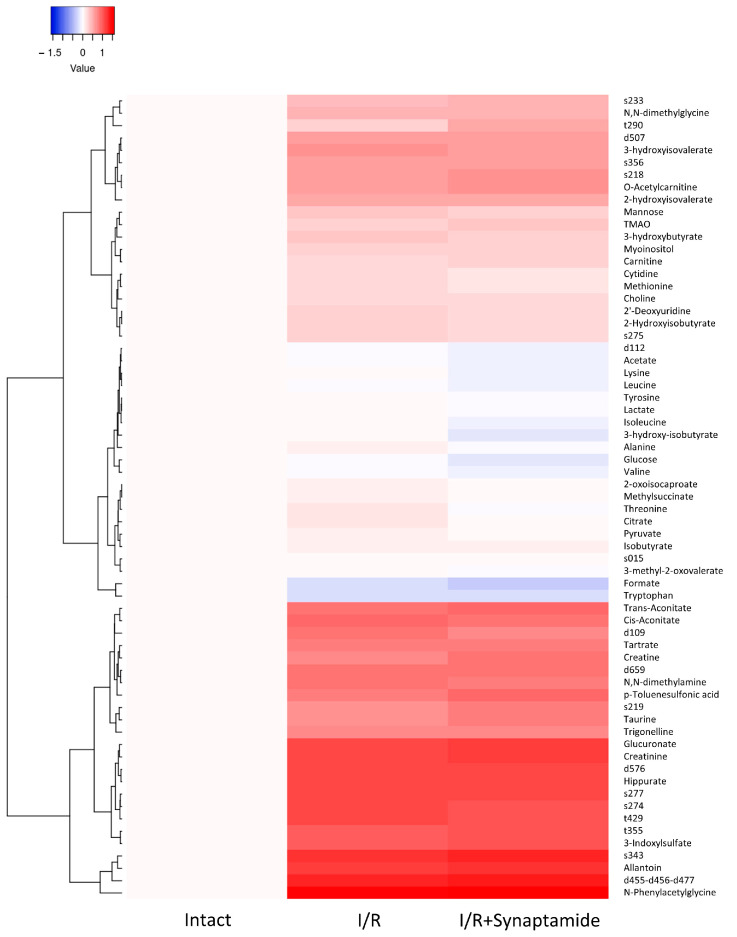
Clustering analysis of metabolic changes in the serum of rats after I/R and I/R with synaptamide administration.

**Figure 8 ijms-25-01500-f008:**
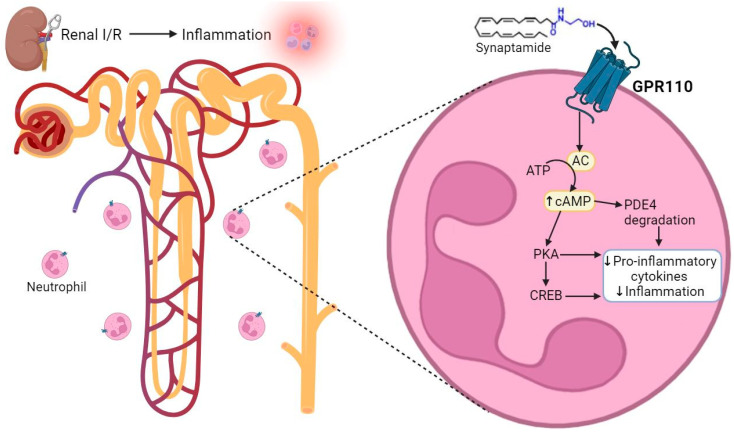
Possible mechanisms of the synaptamide/GPR110-mediated reduction in inflammation during I/R-induced AKI. AC—adenylyl cyclase, ATP—adenosine triphosphate, cAMP—cyclic adenosine monophosphate, CREB—cAMP response element-binding protein, GPR110—G-protein-coupled receptor 110, I/R—ischemia/reperfusion, PKA—protein kinase A, PDE4—Phosphodiesterase-4.

**Figure 9 ijms-25-01500-f009:**
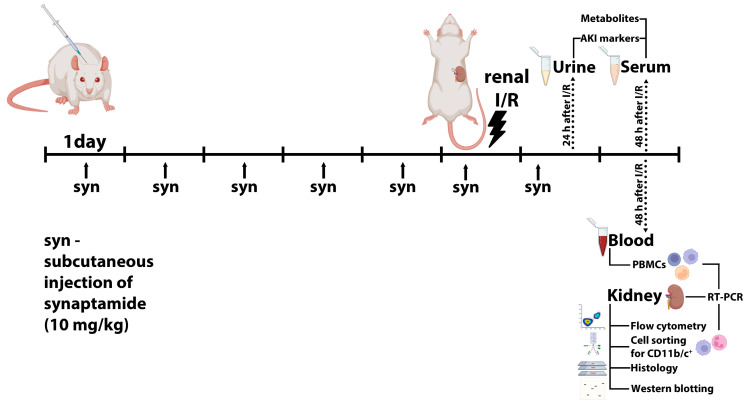
Experimental design. Synaptamide was injected subcutaneously for 5 days before renal I/R, 1 h before I/R, and on the next day after the procedure. Urine, blood, and kidney tissue samples were collected for further analysis.

**Table 1 ijms-25-01500-t001:** Sequences of primers used for gene expression estimation.

Gene Name	The Sequence of Forward Primer 5′- to 3′	The Sequence of Reverse Primer 5′- to 3′
*C3*	AAGCCCAACACCAGCTACATC	ACTTCTGATCCTGGCATTCTTCT
*CD11a*	TGGCAGATGTGGTTGTAGG	TCTGGAAGCACACCTTGAG
*CD45*	AAGCAATACCACCACAAGCACAG	TGGAGTACATGAGCCATTGGAGAG
*CD68*	TTGAACCCGAACAAAACCAAGGTC	GAGAATGTCCACTGTGCTGCTTG
*CD86*	TCAGATGCTGTTCCTGTGAAGAGG	TGAAGTCGTAGAGCCTGGTTATCC
*CD163*	ACAACCGATGCTCAGGAAGAGTAG	CAAGCCAGATTTGTCCAGAACCAG
*CXCL1*	GTGGCAGGGATTCACTTCAAGAAC	GGGACACCCTTTAGCATCTTTTGG
*COX-2*	TACGTGTTGACGTCCAGATC	TGGAGAAAGCTTCCCAACTT
*GPR110*	CATACATAGGGCTGGGCGTCTC	TTGCGTGTGTAGGAGGTTTGGC
*HPRT*	CTCATGGACTGATTATGGACAGGAC	GCAGGTCAGCAAAGAACTTATAGCC
*IL-1b*	CACCTCTCAAGCAGAGCACAG	GGGTTCCATGGTGAAGTCAAC
*Il-6*	CTGGTCTTCTGGAGTTCCGT	TGGTCTTGGTCCTTAGCCAC
*IL10*	GCTGAAGACCCTCTGGATAC	CCAGGCTTACCTTATTAAAATCATTC
*iNOS*	CCACAATAGTACAATACTACTTGG	ACGAGGTGTTCAGCGTGCTCCACG
*KIM-1*	AGGAAGCCGCAGAGAAAC	ATAATGATGTACCTGGTGACAAC
*RPLP0*	CACAGTACCTGCTCAGAACAC	ACCTTGTCTCCAGTCTTTATCAG
*TGFβ1*	CTACGCCAAAGAAGTCACC	CACTGCTTCCCGAATGTC
*TNF-α*	CAAGGAGGAGAAGTTCCCAA	TGATCTGAGTGTGAGGGTCTG
*UBC*	TCGTACCTTTCTCACCACAGTATCTAG	GAAAACTAAGACCCTCCCCATCA

## Data Availability

The data that support the findings of this study are available from the corresponding author upon reasonable request.

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
