# Peer review of "Anti-Inflammatory Effect of Synaptamide in Ischemic Acute Kidney Injury and the Role of G-Protein-Coupled Receptor 110"

_ijms, 2024, doi:10.3390/ijms25031500_

Round 1

Reviewer 1 Report

Comments and Suggestions for Authors

Thank you for the invitation to review the paper entitled: Anti-inflammatory effect of synaptamide in ischemic acute kidney injury and the role of GPR110

 The topic of the work is valid, the purpose is general (should be described in more detail), the choice of methods is correct.

 Results described and discussed correctly.

 I point out a few corrections:

1.       indicate precisely the type of hospital (line 48)

2.       identify examples of products of neutrophils (line 55, 56)

3.       describe the purpose of the work in more detail (lines 75-82)

4.       describe methods (with what sensitivity) for particular compounds (section Materials and Methods)

Reviewer 2 Report

Comments and Suggestions for Authors

Inflammation and oxidative stress are the main pathological processes that accompany acute ischemic injury of kidneys and other organs.
Based on this, these factors are often chosen as targets for treating AKI in various experimental and clinical studies.
In their study, for the first time, authors have demonstrated the anti-inflammatory effect of synaptamide in a model of ischemic AKI.
I believe the study was planned and conducted correctly, and I have no objections.

The aim of the study  - I think it should be more precise because it is a bit missing in the introduction.

I was wondering about the issue of the role of GPR110 - maybe a diagram showing the metabolic pathways and the role of GPR110 would be helpful.
Moreover, the authors mentioned that they identified possible mechanisms by which the therapeutic effect of synaptamide could be achieved in AKI. It would be precious if it were possible to create a diagram showing these possible mechanisms; it would undoubtedly increase the attractiveness of this interesting article.

The conclusions are descriptive, not conclusive; I think this should be slightly modified.

What about future perspectives?
Besides, I think the study's limitations and strengths should be provided in the article.

Comments on the Quality of English Language

Minor editing of English language is required.

Reviewer 3 Report

Comments and Suggestions for Authors

This study by Brezgunova AA et al, entitled Anti-inflammatory effect of synaptamide in ischemic acute kidney injury and the role of GPR110" was designed to investigate the effects of synaptamide on inflammation under AKI induced by renal I/R. Pro-inflammatory mediators, leukocyte populations, GPR110 expression, kidney histology and serum metabolic profile were analyzed. The authors demonstrate that synaptamide treatment attenuated the inflammatory response in renal tissue during ischemic AKI, which is achieved through a GPR110-mediated pathway in neutrophils, and reduced proinflammatory interleukin production.

Here are some additional comments on the manuscript.

(1)    The authors conclude that the anti-inflammatory effects of synaptamide is through GPR110-mediated pathway in neutrophils. Based on the upregulation of GPR110 mRNA expression in the IR+ synaptophysin group, That's insufficient to support that the anti-inflammatory effect is via the GPR110 pathway. By using the GPR110 activators and inhibitors should be considered in trials. In addition, the group of intact+synaptophysin must be included.

(2)    The co-author's publications (J Chem Neuroanat. 2023 Dec:134:102361., Int J Mol Sci. 2023 Mar 27;24(7):6273., Brain Sci. 2021 Nov 26;11(12):1561.) were using 4 mg/kg/day of synaptamide in their reports. Why in this study the authors increase to 10mg/kg as working concentration, what are the judgments about dosage and duration of treatment?

(3)    Figure 1C, 1D, 1E, 1F, 1G, 1H, Supplementary Figure 1A and 1B, 3D, 4D 6A, 6B, 6C, 6D and 7 show only increasing or decreasing trends, and do not reach statistically significant differences in comparisons between the intact group, the IR group, and the IR+synaptamide group. Loading control for Western Blot is missing. 

Round 2

Reviewer 3 Report

Comments and Suggestions for Authors

The authors have answered the reviewer's comments and improved the manuscript. I have no further comments.